Three-dimensional paleohistology of the scale and median fin spine of Lophosteus superbus (Pander 1856)

Jerve Anna ajerve@ic.ac.uk ajerve@gmail.com 1 2
Qu Qingming 1 3
Sanchez Sophie 4 5
Blom Henning 1
Ahlberg Per Erik 1
1 Department of Organismal Biology, Uppsala Universitet , Uppsala , Sweden
2 Department of Life Sciences, Imperial College London , Ascot , United Kingdom
3 Centre for Advanced Research in Environmental Genomics, University of Ottawa , Ottawa , Canada
4 Science of Life Laboratory, Uppsala Universitet , Uppsala , Sweden
5 European Synchrotron Radiation Facility , Grenoble , France
Sues Hans-Dieter
Electronic publication date: 2016 Nov 2
Publication date: 2016
Volume: 4
Electronic Location ID: e2521
Received 2016 Jun 1; Accepted 2016 Sep 2
Copyright: ©2016 Jerve et al.
Copyright year: 2016
Copyright holder: Jerve et al.
License: This is an open access article distributed under the terms of the Creative Commons Attribution License, which permits unrestricted use, distribution, reproduction and adaptation in any medium and for any purpose provided that it is properly attributed. For attribution, the original author(s), title, publication source (PeerJ) and either DOI or URL of the article must be cited.
License URL: https://creativecommons.org/licenses/by/4.0/

Keywords: Paleohistology, Fin spine, Scale, Synchrotron, Lophosteus, Paleontology

Funding: European Synchrotron Radiation Facility proposal EC 688 Swedish Research Council Knut and Alice Wallenberg Foundation ERC Advanced Investigator Grant 233111 Vetenskapsrådet The European Synchrotron Radiation Facility proposal EC 688 funded the scanning of the material. All other funding was provided by the Swedish Research Council awarded to HB, a Wallenberg Scholarship from the Knut and Alice Wallenberg Foundation awarded to PEA. SS was supported by an ERC Advanced Investigator Grant 233111 awarded to PEA, as well as the aforementioned Wallenberg Scholarship. QQ is supported by an International Postdoc Grant from Vetenskapsrådet. The funders had no role in study design, data collection and analysis, decision to publish, or preparation of the manuscript.

==============================
Lophosteus superbus is one of only a handful of probable stem-group osteichthyans known from the fossil record. First collected and described in the late 19th century from the upper Silurian Saaremaa Cliff locality in Estonia, it is known from a wealth of disarticulated scales, fin spines, and bone fragments. In this study we provide the first description of the morphology and paleohistology of a fin spine and scale from Lophosteus using virtual thin sections and 3D reconstructions that were segmented using phase-contrast synchrotron X-ray microtomography. These data reveal that both structures have fully or partially buried odontodes, which retain fine morphological details in older generations, including sharp nodes and serrated ridgelets. The vascular architecture of the fin spine tip, which is composed of several layers of longitudinally directed bone vascular canals, is much more complex compared to the bulbous horizontal canals within the scale, but they both have distinctive networks of ascending canals within each individual odontode. Other histological characteristics that can be observed from the data are cell spaces and Sharpey’s fibers that, when combined with the vascularization, could help to provide insights into the growth of the structure. The 3D data of the scales from Lophosteus superbus is similar to comparable data from other fossil osteichthyans, and the morphology of the reconstructed buried odontodes from this species is identical to scale material of Lophosteus ohesaarensis, casting doubt on the validity of that species. The 3D data presented in this paper is the first for fossil fin spines and so comparable data is not yet available. However, the overall morphology and histology seems to be similar to the structure of placoderm dermal plates. The 3D datasets presented here provide show that microtomography is a powerful tool for investigating the three-dimensional microstructure of fossils, which is difficult to study using traditional histological methods. These results also increase the utility of fin spines and scales suggest that these data are a potentially rich source of morphological data that could be used for studying questions relating to early vertebrate growth and evolution.

Introduction

Research into the early evolution of gnathostomes (jawed vertebrates) is currently undergoing a paradigm shift, with far-reaching effects including a renewed interest in the enigmatic fossil fish Lophosteus from the Late Silurian of Estonia. For many decades, virtually all research in the field has incorporated the assumption that the macromeric dermal bone skeleton of osteichthyans (extant bony fishes and tetrapods), that is their stable and historically conserved pattern of named bones such as maxilla and dentary, evolved directly from a micromeric ancestral condition consisting of scales or small tesserae without individual identities (Janvier, 1996). The similarly macromeric dermal skeleton of placoderms (jawed, armored stem-gnathostomes of the Silurian and Devonian periods) was deemed to have an independent origin from a micromeric ancestor, and any pattern matches between the placoderm and osteichthyan skeletons were interpreted as convergent. Recently, it has become clear that this hypothesis is untenable: the discovery of placoderm-like characters in the dermal skeletons of the earliest osteichthyans (Zhu, Yu & Janvier, 1999; Zhu et al., 2009; Zhu et al., 2012), and in particular the Silurian “maxillate placoderm” Entelognathus which combines a full set of osteichthyan marginal jaw bones with an otherwise typical placoderm skeleton (Zhu et al., 2013), has demonstrated that macromery is homologous in osteichthyans and placoderms. Current consensus is that jawed vertebrates primitively have macromeric dermal skeletons, as shown by placoderms, and that this condition is retained in osteichthyans but lost in acanthodians (“spiny sharks”, a Silurian to Permian group of jawed fishes) and chondrichthyans (extant cartilaginous fishes) which have become micromeric (Zhu et al., 2013; Dupret et al., 2014).

This new consensus casts a spotlight on the few macromeric fossil taxa that appear to bridge the—still quite substantial—morphological gap between placoderms and osteichthyans. These forms, which have the potential to illuminate the origin of the gnathostome crown group, include Janusiscus (Giles, Friedman & Brazeau, 2015), Dialipina (Schultze & Cumbaa, 2001), Ligulalepis (Basden & Young, 2001), Andreolepis (Janvier, 1978; Botella et al., 2007; Qu et al., 2013) and Lophosteus (Pander, 1856; Gross, 1969; Gross, 1971; Botella et al., 2007), all from the Late Silurian to Early Devonian. Janusiscus is currently interpreted as a crownward stem gnathostome (Giles, Rücklin & Donoghue, 2013), the others as stem osteichthyans or unresolved basal osteichthyans (Botella et al., 2007; Zhu et al., 2013; Giles, Friedman & Brazeau, 2015). While the first three genera are known from complete specimens (Dialipina) or braincases with attached skull roofs (Janusiscus, Ligulalepis), Andreolepis and Lophosteus are represented only by disarticulated fragments and occasional complete bones from the dermal skeleton. However, they compensate for this by the abundance of the material and in particular by the superb histological preservation of the bones (Gross, 1969; Gross, 1971; Qu et al., 2013). This enables us to investigate the tissue organization and growth modes of their dermal skeletons, uncovering a rich source not only of paleobiological information but also of phylogenetically informative characters. The potential value of the histological data set has been greatly enhanced in recent years by the application of propagation phase contrast synchrotron microtomography (PPC-SRµCT), which allows us to visualize the histology non-destructively in three dimensions with single-cell resolution (Sanchez et al., 2012). We present here the first PPC-SRµCT investigation of the scales and dermal fin spines of Lophosteus.

The scales and spines of Lophosteus superbus are among the most abundant remains collected from Ohessaare Cliff on the island of Saaremaa in Estonia since Pander first described this taxon in 1856. Gross (1969) and Gross (1971) provided the most detailed description of L. superbus, which he based on an assemblage collected from the same locality. Since then several other species of Lophosteus have been described from across the globe, including localities in North America (Märss et al., 1998), Australia (Burrow, 1995), and central and eastern Europe (Märss, 1997; Botella et al., 2007; Cunningham et al., 2012) indicating that Lophosteus was widely distributed. For a more comprehensive overview of Lophosteus systematics, see Schultze & Märss (2004).

Because our knowledge of Lophosteus is based on bone fragments, scales, and spines, it has been difficult to determine precisely where it fits into the larger picture of early gnathostome evolution. However, after careful examination of the morphology and histology of the material, Gross (1969) was able to confirm that the disarticulated scales, spines and dermal bones from Ohessaare attributed to Lophosteus do indeed belong to one genus. Lophosteus is currently considered by some to be a stem-osteichthyan (Botella et al., 2007; Cunningham et al., 2012), but it has also been proposed as sharing affinities with crown osteichthyans (Gross, 1971; Rohon, 1893), acanthodians (Schultze & Märss, 2004), and placoderms (Burrow, 1995). The view that Lophosteus is the least crownward stem-osteichthyan (Botella et al., 2007) has a significant impact on interpreting gnathostome phylogeny and the acquisition of crown gnathostome characteristics (Brazeau, 2009; Brazeau & Friedman, 2014; Giles, Friedman & Brazeau, 2015; Qu et al., 2015b).

The histology of scales and spines of early gnathostomes can reveal important information relating to the development and evolution of these structures and of the hard tissues that form them, as well as the phylogenetic relationships of the animals that carried them (Ørvig, 1951; Ørvig, 1977; Burrow & Turner, 1999; Valiukevičius & Burrow, 2005; Schultze, 2015; Giles, Rücklin & Donoghue, 2013). For example, Lophosteus and Andreolepis (a slightly earlier Late Silurian taxon from Gotland, Sweden) were once considered to be closely related and were grouped in the family Lophosteidae (Gross, 1969; Gross, 1971; Schultze & Märss, 2004). However, the presence of enamel in Andreolepis scales and the absence of this tissue in Lophosteus (Gross, 1969) contributed to altering this view (Otto, 1991; Cunningham et al., 2012; Qu et al., 2015b). Given the current lack of articulated material, the scales and spines of Lophosteus are the most readily available source of data for phylogenetically important characters. Other bones have been identified, including jawbones (Botella et al., 2007; Cunningham et al., 2012), and a new extensive material of cranial and postcranial dermal bones from Ohessaare is currently under study (Ahlberg et al., 2013). The specimens presented here derive from this new material.

The bone histology of Lophosteus has been described from two-dimensional (2D) thin sections (Gross, 1969; Gross, 1971; Märss, 1986; Burrow, 1995) but the three-dimensional (3D) histological arrangement of the spine and scale has never been investigated. In this paper we present detailed 3D descriptions from a fin spine and a scale of Lophosteus superbus, based on PPC-SRµCT scans made at the European Synchrotron Radiation Facility (ESRF) in Grenoble, France. These data increase our understanding of the development of the spines and scales from this species and permit us to discuss potential new paleohistological characters, which will become crucial for future phylogenetic analyses.

Materials and Methods

Specimens

The material from Lophosteus superbus that is described in this paper was collected as part of a larger effort to amass material of Paleozoic vertebrates from Ohesaare Cliff in Estonia by Uppsala University, Sweden, and the Institute of Geology at the Tallinn University of Technology (GIT), Estonia. The material was collected from the upper Pridoli Ohesaare Cliff beds (Žigaitė et al., 2015) in large limestone blocks that were chemically prepared using a weak solution of acetic acid in water (pH 3.65) at the fossil preparation laboratory at Lund University, Sweden. Published material from this project is held at the GIT in Estonia and the rest of the material is housed at the Evolution Museum at Uppsala University, Sweden. 3D printed models of both specimens have been catalogued in the collection.

The depth-length ratio (>1.5) of the scanned scale (GIT 727-1) indicates it is probably a central or anterior trunk scale based on comparison to squamation of other early osteichthyans (Chen et al., 2012; Jessen, 1968; Qu, Zhu & Wang, 2013; Trinajstic, 1999). The scan of the scale is incomplete, missing the most dorsal and ventral part (Fig. 1). The Lophosteus fin spine (GIT 727-2) scan is also incomplete and only includes the most distal part of the structure. However, this does not affect the study of its general growth pattern and 3D architecture (see ‘Results’). The synchrotron data will be made available through the ESRF palaeontology database (http://paleo.esrf.eu).

Figure 1 The scanned scale (GIT 727-1) of Lophosteus superbus.

Scale in crown (A), basal (B) and anterior (C) view. Red and green lines in (C) mark the cutting planes for the virtual thin sections in (D), (E); Blue line in (C) marks the cutting plane for the virtual thin section in (F). (D) Vertical anteroposterior virtual thin section showing the embedded odontode (eo) and other histological structures. Arrow head marks the same loose region in (D), (E) and (F). (E) Vertical dorsoventral virtual thin section showing the continuous loose region in the middle of the bony base. (F) Horizontal virtual thin section. (G) Zoom-in of a region between two crown ridges of (GIT 727-1) showing bone-like tissue and osteocyte-like spaces. (H) Zoom-in of a region surrounded by crown ridges of a Romundina stellina scale (NRM-PZ P.15952), rendered in VG Studio MAX 2.2 using data from Rücklin & Donoghue (2015).

Synchrotron parameters

Both the scale and the spine of Lophosteus (GIT 727-1 and GIT 727-2) were imaged using Propagation Phase-Contrast Synchrotron X-ray Microtomography (PPC-SRµCT) at beamline ID19 of the European Synchrotron Radiation Facility (ESRF), France. The samples were scanned with the energy of 30 keV in monochromatic conditions, using a single crystal 2.5 nm period W/B4C multilayer monochromator. The beam was filtered with 2 mm of aluminum. The insertion device used was a U32 undulator with a gap of 12.38 mm. The detector was a FreLoN 2K14 CCD camera (Labiche et al., 2007). In association with the microscope optic and a 10 µm-thick gadolinium gallium garnet (GGG) scintillator doped with europium (Martin et al., 2009), the camera provided an isotropic voxel size of 0.678 µm. The samples were fixed at a propagation distance of 30 mm from the detector. Two thousand projections were performed during continuous rotation over 180 degrees. The time of exposure per projection was of 0.3 s. The field of view at high resolution was restricted to 1.4 mm, and therefore only specific regions of the scale and spine were imaged. The data obtained in edge detection mode were reconstructed using a classical filtered back-projection algorithm (PyHST software, ESRF). Acquisition artifacts—such as ring artifacts—were filtered while processing the data. Segmentation and rendering were done using the software VG StudioMax 2.2 (Volume Graphics, Heidelberg), following the protocols established by Qu et al. (2015a).

Terminology

The terminology from Gross (1969) and Gross (1971) forms the basis of our description, and most of the terms describing 2D histology are adopted in 3D data. Our description of ornament and morphology follows the terminology established by Schultze & Märss (2004). General histology terms regarding vertebrate hard tissues follow Francillon-Vieillot et al. (1990). The homology assessment of the canal system in 3D follows previous work on the Psarolepis and Andreolepis scales (Qu et al., 2016).

Results

The external morphology of the Lophosteus scale (GIT 727-1) and spine tip (GIT 727-2) described here are identical with the scales and symmetrical spines that are figured and described in Gross (1969). The spine also shares characteristics with the median dorsal spine described by Otto (1991). These similarities include the overall shape, organization of tubercles, cross-sectional shape, morphology of the posterior surface of the spine and the histological arrangement into different tissue layers (Gross, 1969; Otto, 1991).

Scale morphology and 3D histology

Composition and overall morphology

The scale is rhombic in shape in crown view, with a broad anterior overlapped field (Fig. 1A). There is a broad keel at the center of the scale in basal view, accompanied by an anterior ledge and posterior ledge. Numerous pores are present on the keel (Fig. 1B). Morphologically the scanned scale conforms to the large Lophosteus scales as described by Gross (1969).

The histology of the scale can be subdivided into three layers, as originally recognized by Gross (1969): a basal bony layer with numerous osteocyte lacunae, a middle layer with a horizontal vasculature and osteocytes and a top layer made of dentine mainly (ornament). Besides the three main basal canals connecting with the horizontal vascular canals, there are several isolated canals in the basal layer (Figs. 1B and 1D). No secondary bone deposition occurred to form osteon-like structures around these basal canals, and the bony base is made of pseudo-lamellar bone (Gross, 1969; pers. obs. by Q Qu (2015) based on classical thin sections). There is a peculiar loosely textured region within the basal bony layer, and this region is continuous along the length of anterior overlapped field (Figs. 1D and 1E). This loosely textured region seems to have been occupied by numerous fibers in vivo, and Sharpey’s fibers extend from this region to the basal surface of the scale (Fig. 1E). The ornament denticles are comprised of dentine, which gradually change to cellular bone basally (Fig. 1D). However, there is no clear boundary between dentine and bone. The dentine of large younger odontodes becomes more complicated, with regular dentine tubules in the outer layer and denteons in the inner layer (Fig. 2B). These denteons have a central ascending canal from which thin tubules radiate, very similar to primary osteons in bony tissue, but are composed of dentine (Fig. 2B).

Figure 2 Histological detail of a large odontode of (GIT 727-1).

(A) Posteroventral view of the scale crown, showing the cutting plane of the virtual thin section in (B), (C). (B) Horizontal virtual thin section cutting through a large dental ridge of the crown, with default contrast setup in VGStudio MAX 2.2. (C) The same section as in (B) but the image contrast is increased in VGStudio MAX 2.2 to show denteons in a large odontode.

Ornamentation

Although the scan is incomplete, missing a small dorsal portion and a ventral portion of the scale (Fig. 1A), it is possible to reconstruct the growth history of the scale crown by rendering the embedded odontodes (Fig. 3). Four generations of odontodes have been identified. Each generation consists of multiple odontodes, which share the same bony base (Fig. 3A) and are similar in size and shape (Fig. 3B). Odontodes of each generation form a continuous sheet and are connected by bony tissues. Odontodes of a younger generation never cover the previous generation odontodes completely, conforming to an areal growth pattern.

Figure 3 Segmentation of the virtual thin sections of the Lophosteus scale and the growth history of the scale crown.

(A) Vertical dorsoventral virtual thin section showing embedded odontodes in the crown (B) and their surfaces selected in VG VGStudio MAX 2.2. (C) Rendered odontodes in sequential order showing the growth history of the scale, crown view. Red line in (C) marks the cutting plan of the virtual thin section in (A).

Figure 4 Growth history of the scale crown in posteroventral view.

Arrowheads mark the small nodules on the ridgelets of embedded odontodes.

First generation odontodes are triangular in crown view, with two major (middle) ridgelets converging to the posterior tip on each odontode (Fig. 3C). There is a long ridgelet with nodular serrations on the ventral side of each odontode (Fig. 4A), but the number of nodules varies. Second generation odontodes are larger and more elongated than first generation odontodes. There also are more ridgelets on each odontode, with dorsal and ventral ridgelets having nodular serrations (Figs. 3C and 4B). Nodules become more prominent basally on each ridgelet. In posterior view each odontode is stellate-shaped with ridgelets radiating from the posterior tip (Fig. 4B). Third and fourth generation odontodes are larger than older odontodes. Their posterior tips become blunt, suggesting strong postmortem erosion. There are more ridgelets on these odontodes than on older odontodes. Nodular serrations on embedded part of ridgelets are clearly visible, while remnants of nodules after erosion become faint on exposed surfaces (Fig. 4C).

Vascularization

The whole canal system is subdivided into three parts: 1. A basal part with vertical basal canals in the bony base (Fig. 5, yellow); 2. A middle part with horizontal canals lying below odontodes (Fig. 5, pink and green); 3. A crown part with vertical ascending canals (surrounded by denteons) lying within odontodes (Fig. 5B, bright red). This division is consistent with the three layers of 2D histology of the scale. The middle horizontal canals have numerous openings (Fig. 5, green) on the surface of the scale.

Figure 5 Three-dimensional vasculature of the Lophosteus scale.

(A–E) Crown view and (F, G) basal view. Four generations of odontodes are rendered transparent to show their underlying vascular canals in (B–E). The first generation of odontodes is shown in (G), with all basal canals below these odontodes. (H, I) Posterolateral view with rendered surfaces of odontodes.

Figure 6 3D renderings of the tip of the fin spine of Lophosteus superbus (GIT 727-2) in (A) dorsal, (B) left lateral, (C) ventral, and (D) right lateral views. Scale bar for (A–D) is 300 µm. (E) Magnified view of a portion of the left lateral view of the spine, indicated in (B) by the red box. Scale bar is 100 µm. (F) Virtual thin section in transverse view of the fin spine indicated by the red line in B, which highlights its general histological features. The color coding denotes zones (boundaries are approximations) of the spine tip, including bone with fewer bone cells spaces (blue), bone with many cell spaces (purple), dentine (red), and Sharpey’s fibers (green). Scale bar is 200 µm.

There are three basal canals connecting with the middle horizontal canals (Fig. 5), but several isolated basal canals that do not connect with other canals are also present (bco.n in Fig. 1B). The middle horizontal canal system consists of bulging sack-like cavities joined together by much narrower canals that show semi-regular spacing (Fig. 5A). Except where overgrown by later odontodes, these narrow canals (green) open onto the external surface of the scale through a ring of foramina around the base of each odontode (Fig. 1E). Since the ascending canals (Fig. 5, bright red) originate from the middle canals and connect to terminal dentine tubules within odontodes, the middle horizontal canals are considered as vascular canals too. Each odontode overlies a sack-like cavity of the middle horizontal canal system, and younger larger odontodes have correspondingly larger horizontal cavities below them (Figs. 5B–5E). Within the odontodes some ascending canals connect with each other by forming arcade-like structures (“Arkadenkanal” in Gross, 1969) (adc, Fig. 5). Dentine tubules originate from the ascending canals and arcade canals that should thus be considered as pulp canals proper or cavities.

Spine tip morphology and 3D histology

Composition and overall morphology

The spine is constructed of dermal bone that is covered by dentine odontodes (Figs. 6A–6D). The lateral sides of the spine are ornamented in asymmetrical stellate tubercles, or odontodes, that are ornamented with “ridgelets”, as described in Märss et al. (1998) and Schultze & Märss (2004) (Fig. 6E). Odontodes can be partially overlapping or freestanding with the former being smaller and shorter than those that are freestanding (Fig. 6B; compare Figs. 7E–7F and 8H–8K to 8L–8N). There are many pores that represent vascular canal openings located on the surface of the spine, usually between odontodes on the lateral sides of the spine (Figs. 6E and 6F; vco), and parallel with the edges of the posterior surface (ps) of the spine (Fig. 6C). They are typically not as regularly arranged, or as closely associated with individual odontodes, as the corresponding pores on the scales. Similar to the observations of Otto (1991) and Gross (1969), we note that the leading edge of the spine is constructed of a linear row of slightly offset, elongated and unornamented odontodes (Figs. 6A–6B, 6D and 6F; leo). The posterior surface of the spine is constructed of bone, bears no odontodes, and is narrower than the lateral sides (Figs. 6A–6D). The bony part of the spine along its posterior surface is slightly concave with short ridges running along the sides (Fig. 6C). The ridges fade and flatten out distally and the surface narrows to a rounded boundary, which demarcates the bone from a dentine tubercle that comprises a portion of the tip of the spine (Figs. 6C–6D), along with the odontode at the spine tip’s leading edge (between the lateral faces of the spine) (Fig. 6D).

Figure 7 Virtual thin sections of lateroventral surface of the spine of L. superbus (GIT 727-2) showing (A) the morphology of the area and (B) highlighting first and second generation odontodes. (A) and (B) are the same image and share the same coordinate system and scale bar. Scale bar is 80 µm. (C) Rendered surface of the entire tip of the fin spine showing the location in red of the (A) and (B) virtual thin sections. The pink portion of the surface shows the location of the partially buried odontodes. (D) separates the pink region from the rest of the 3D rendering while (E) isolates the first-generation odontodes and buried surface (dark pink) and (F) shows the position and morphology of second-generation odontodes (light pink). (C–F) share coordinate system and scale bars. Scale bars are 200 µm.

Figure 8 Virtual thin sections of the Lophosteus (GIT 727-2) spine taken in (A) frontal and (B) sagittal planes and a (C) 3D rendering of the entire area with odontodes (D–Q) marked to show position to each other.

Scale bars are as follows: (A) is 150 µm, (B) is 200 µm, and (C) is 250 µm. 3D renderings of individual first-generation odontodes to show the morphology and ornamentation in (D) dorsal, (E, F) lateral, and (G) oblique lateral views. (H–K) illustrates the same, but with a different first-generation odontode. Yellow arrow indicates the buried denticles on first generation odontodes. Scale bars for (D)–(K) are 75 µ. (L) shows the morphology of a second-generation odontode in dorsal view, in addition to (M) lateral and (N) oblique lateral views. Scale bars for L and N are 150 µm and (M) is 100 µm. (O–Q) illustrates the morphology of another second-generation odontode in the same orientation, as the odontode figured in (L)–(N). Scale bars are 150 µm.

Ornamentation

Overlapping odontodes like those noted by Gross (1969) and Otto (1991) can be seen on the scan data and have been reconstructed here (Figs. 7 and 8). Individual odontodes are elongate with a ridge that extends the length of each structure. Ridglets extend toward the base of the odontodes and from the median surface of the structure to form an overall stellate pattern. These odontodes are classified as either first- or second-generation as their relative ages can be determined (Figs. 7 and 8). Second generation odontodes (sgo) are younger and they share the outer bone surface as a depositional boundary (Fig. 7). This ornamentation tends to be larger and longer with broad, smooth surfaces and ridgelets, relative to the older first generation odontodes (Figs. 8L–8Q) (fgo) that are partially buried (Fig. 7). First generation odontodes share a depositional surface under the bony surface of the spine (Figs. 7A–7B and 7E). The buried first generation odontodes are ornamented with a series of diagonally crossing sharp ridgelets of different lengths (Figs. 8D–8G; 8H–8K). A series of tooth-shaped nodules that form rough serrations can be seen on the distal end of these tubercles (Figs. 8H–8K; yellow arrow).

Overall histology

Gross (1969) and Otto (1991) described the 2D histology of the spine, which is similar to that of the scale, including a lamellar bony layer that bears longitudinal canals, a middle “spongy” layer, and an ornamented dentine layer. Here, we identify a more compact bone with fewer bone cell spaces (Fig. 6F; blue), a layer of bone with many bone cell spaces, pseudocanals, and void spaces (Fig. 6F; purple), and an outer ornamented dentine layer (Fig. 6F; red). Additionally, the bony posterior region of the spine contains numerous Sharpey’s fibers for attachment with the fin (Fig. 6F; green). The dentine layer is not continuous over the surface of the spine.

Figure 9 Breakdown of the vascularization within the tip of the fin spine of Lophosteus (GIT 727-2).

(A) and (B) are cross-sections of 3D renderings and scan data of slices 472–680 and 786–984 (thin sections are slices 580 and 880), respectively, to show how each layer of canals relates to each other. Orange, central vascular canal, cvc; Light blue, 1st tier canals, 1st; Dark Blue, median canal, mc; Yellow, basal canal, bc; Purple, 2nd tier canals, 2nd; Pink, 3rd tier canals, 3rd. 3D renderings of the surface of the spine also included in white at the end of each vascular canal. Color scheme is the same throughout figure. Scale bars for (A) and (B) are 250 µm. (C) left lateral and (D) posterior views. The locations of the virtual thin sections and renderings are labeled in red on the surface rendering of the spine in (C). Scale bars for (C) and (D) are 300 µm. Yellow arrows explained in text.

Figure 10 Virtual thin sections and 3D renderings of odontode morphology and vascularization in (A) sagittal view of the network of ascending canals within a single odontode and (B) and (C) illustrate the location of the ascending canals (red) in relation to the outer layer of vascular canals (pink). Scales bars are 200 µm. (D) is a 3D rendering of the ascending canals (red) and the outer tier of canals of the spine (pink) in a second-generation odontode. (E) shows the semi-transparent surface of the same odontode and (F) shows the full surface of the odontode. Scale bars are 100 µm. (G–I) shows the same as (D–F), but in two-first-generation odontodes. Scale bars are 100 µm. Yellow arrows explained in text.

Vascularization

Vascular canals (vc) within the body of the spine surround a large central vascular canal (cvc), which is non-uniform and roughly triangular in cross-section and appears to be composed of several “lobes” (Fig. 9). Canals which we designate as bone vascular canals (Fig. 9A; bvc1, teal/blue) bifurcate from the central vascular canal (Fig. 9A; cvc, orange) and are associated with the main bony core of the spine. Near the central vascular canal, this layer consists of long and narrow canals that are closely situated together. Distally, these canals become smaller in size and are more rounded in cross-section (Fig. 9A). Bone vascular canals meet posteriorly at a large median canal (Fig. 9A; mc, purple) that runs longitudinally along the length of the central vascular canal. The median canal is the second largest component of the vascular network, with many laterally bifurcating arms. Large bone vascular canals connect the median canal to the central vascular canal anteriorly in places but these can also be connected directly to each other (Figs. 9A and 9B). Smaller canals connect the median canal posteriorly to the basal canal (Fig. 9A; bc, yellow). The basal canal is narrower than the median canal and also runs the length of the central vascular canal (Figs. 9B–9D). At the distal end of the spine tip this canal bifurcates into several smaller canals that form some of the vascular network of the tip of the spine (Figs. 9C and 9D; pink; yellow arrows). The second layer of bone vascular canals (Fig. 9; dvc2, pink), are outermost in position and are associated with the outermost layer of bone that has many cell and void spaces (Fig. 9B). They typically have a clear boundary with the rest of the vascular network, which is marked by short, thin canals (Figs. 9A and 9B). The vascular network of the most distal part of the tip of the spine is continuous with the vascular network of the spine body, but all of the layers converge leaving no clear boundaries (Figs. 9C–9D).

Figure 11 (A) Synchrotron scan slice through the Lophosteus fin spine showing the location and organization of different cell and void spaces. Scale bar is 200 µm. Modeled cell spaces include (B) osteocytes and (C–G) unidentifiable void spaces. Scale bars are 45 µm.

Figure 12 (A–C) Synchrotron scan slices of the Lophosteus spine to show the location, morphology, and organization of a layer of void spaces (pseudo-canals?). (D) 3D rendering of the void spaces/psuedocanals and how they fit together with the rest of the vascularization in (E) lateral view of the entire specimen and in (F) cross-section. Scale bars for (A–E) are 250 µm and scale are for (F) is 150 µm.

Some odontodes contain their own smaller vascular network, composed of what are designated here as ascending canals (Fig. 10; adc). Ascending canals form a well-developed, complex network of looping vascular canals involved with the deposition of each odontode. The canals of the network attach to the outer layer of bone vascular canals basally to create a loop distally, whose height reflects the overall morphology of each individual odontode (Figs. 10H and 10I). Branching dentine tubules can be reconstructed and can be observed on the most distal parts of the loops (Figs. 10D and 10G; marked by yellow arrows). The details of the ascending canals are best observed in second-generation odontodes (younger odontodes), because these canals are large and fully open unlike those from first-generation canals, which are small to non-existent and appear to have been secondarily in-filled with dentine (compare Figs. 10D–10E).

Cell distribution

Bone cell spaces are dispersed around the vascular canals and throughout the bony part of the spine (Figs. 11A and 11B). There appears to be a lower density of cell spaces immediately proximal to the central vascular canal in virtual thin sections of the data (Fig. 11A). Clusters of what appear in cross-section to be bone cell spaces are also present and are all located around the outer 2/3 of the bone layer, but these are difficult to identify as such when rendered in 3D (Figs. 11A and 11C–11G) Some of the more enigmatic spaces resemble vascular canals with pocked surfaces (Figs. 11C–11E and 11G), while others render as unidentifiable voids (Fig. 11F). The latter are most likely fiber bundles.

Irregularly shaped canals, referred to as void spaces/pseudo-canals (v) here, are also located in ring around the outer 2/3 of the bone layer (Fig. 12). In section images the pseudo-canals look like vascular canals (Figs. 12A–12C), but when they are rendered in 3D they appear to be irregularly shaped, mostly flat, and do not seem to connect to any of the vascular canals (Figs. 12D–12F). In some respects, they resemble the large clusters of unidentifiable cell spaces mentioned above (Figs. 11C and 11G). The pseudo-canals appear to be located in one layer in the bone and are seen regularly throughout the scan of the spine (Figs. 12D and 12E).

Attachment fibers

Sharpey’s fibers are present in the scan data of the posterior surface of the spine (Fig. 6; shf, green & Fig. 13). In cross-section these are more closely spaced together than the cell spaces in Fig. 11, and they are limited to the area around the posterior surface of the fin spine (Figs. 13A–13D). The fibers are elongated and intersect and exit the surface of the spine at an angle (Figs. 13C–13G). They are difficult to segment and are often not clearly separable from cell spaces.

Figure 13 (A) 3D rendering of the ventral surface of the Lophosteus spine. Red lines indicating the location of the slices for (B) and (C). Scale bar 300 µm. The location of Sharpey’s fibers in the ventral portion of the spine in (B) transverse and (C) sagittal sections. Fibers are highlights in purple in (C). Scale bar for (B) is 200 µm and (C) is 100 µm. (D) Transverse posterior view of the fin spine to illustrate the position of the Sharpey’s fibers. Scale bar is 350 µm. 3D renderings of the Sharpey’s fibers in (E) ventral view and (F) transverse posterior view. Scale bar for (E) is 80 µm and (F) is 70 µm. (G) Cross-section of (F) with semi-transparent surface. Location of (E–G) indicated by the red boxes in (A) and (D). Scale bar is 70 µm.

Discussion

Assigning the material to Lophosteus superbus

Lophosteus superbus was originally diagnosed by Gross (1969) from scales, fin spines, and other bony fragments, all which bear round or elongated stellate odontodes. The scale and spine material described here matches the description provided by Gross (1969) and Schultze & Märss (2004). The morphology of the scale is rhombic and it is ornamented with a series of obliquely placed set of overlapping ridges that do not form a continuous layer of dentine layer (Gross, 1969; Schultze & Märss, 2004). The results of the 3D data presented here have some implications on the taxonomic status of the species Lophosteus ohesaarensis, described from the Ohesaare Cliff locality by Schultze & Märss (2004). L. ohesaarensis was distinguished from L. superbus by the morphology of the ridglets that comprise the individual odontodes on the scales. Schultze & Märss (2004) provided three diagnostic characters for L. ohesaarensis, including, 1. scales with fine parallel ridgelets on crest, which is the highest line of the ridge, ridgelets change angle from 10° to nearly 90° to crest on lateral sides of the ridges; 2. lower part of ridgelets with nodular serrations; 3. anterior overlapped field weakly pustulate. The first two characters are clearly visible in the second-generation odontodes of the described scale (Figs. 3C and 4B). Regarding the third character, the anterior overlapped field is less pustulate in the young scale with the first generation odontodes (Fig. 3C) compared to the mature scale (Fig. 3C). The scales described as L. ohesaarensis are generally smaller with less prominent anterior overlapped field, which suggests that they are most probably juvenile scales of L. superbus. Gross (1969) also has described several scales with small anterior overlapped fields. It is thus more likely that there is only one valid species of Lophosteus from the Ohesaare Cliff locality. However, a similar examination of different ontogenetic stages of L. superbus and L. ohesaarensis material collected from the Ventspils borehole, Latvia, is necessary to confirm their taxonomic status.

Up to eight types of spines and spine-like elements have been attributed to Lophosteus, including symmetrical and asymmetrical forms that are associated with median and paired fins, respectively. The original diagnosis of L. superbus made by Gross (1969) includes a symmetrical spine that is triangular in cross-section with parallel ridges of stellate odontodes that meet at the leading edge. Symmetrical spines recovered from Saaremaa have a similar odontode arrangement, in addition to having rows of ridge-like tubercles that are in parallel row on the lateral parts of the spine and have a long base (Schultze & Märss, 2004). These descriptions differ from the median dorsal spine presented by Otto (1991), who reported that odontodes on the lateral sides of the spine were of varying sizes and not arranged in distinct parallel rows. Otto (1991) noted that leading edge is composed of ridge-like odontodes that are arranged into a single linear row, which was also figured by Gross (1969). It is challenging to assign the spine tip described here to one type of spine because the scan data only represents the most apical region of the structure, but some comparisons can be made (Fig. 6). The spine tip shares a large number of similarities with the symmetrical spines and median dorsal spine of L. superbus material already described by Gross (1969), Otto (1991) and Schultze & Märss (2004). The posterior surface of the spine tip is unornamented and slightly concave proximally and slightly convex to flat at the tip, which is similar to the same surfaces in L. superbus spines described by Gross (1969) and Schultze & Märss (2004). Moreover, the scan data show the multiple generations of odontode growth that are included in the original descriptions by Gross (1969) and the later description by Otto (1991). Overall, the spine tip described in this paper is most similar to the median dorsal spines described by Otto (1991) and Gross (1969) on the basis on the arrangement of the ridge-like odontodes comprising the leading edge. However, there are no data for the proximal part of this spine and so it is difficult to say this with complete certainty.

Lophosteus spine and scale histology comparison

The 3D reconstructions of the current data have revealed several new characters that are shared between the scale and the spine of Lophosteus that otherwise might not have been identified through traditional investigation. Datasets confirm the earlier descriptions made by Gross (1969) with the spine and scale being constructed similarly, with a loosely calcified bony base that is covered in dentine odontodes (Figs. 1 and 6). Within the bony tissue we have identified large void spaces that most likely represent clusters of fibers (Figs. 1D–1F and 11). We also validate the claim that the surface of each structure is not covered by a continuous layer of dentine (Gross, 1969); rather, bony surfaces can be identified between each odontode (Figs. 1G and 6E). The exposed bony surfaces of the scale and spine seem to bear osteoblast spaces, while osteocyte lacunae can be identified from virtual thin sections. The osteocyte spaces are uniquely large in size and may help as a diagnostic tool in future studies (Figs. 7A and 11). Another feature that we can confirm from these data is the presence of a large amount of Sharpey’s fibers connected to the ventral surfaces of both the spine and the scale (Fig. 1, 6F and 13).

Our data further agree with Gross (1969) and Otto (1991) with regard to the presence of overlapping odontodes. The overall morphology of each odontode—elongate, stellate, and ridged—is the same on both scale and spine (Figs. 3 and 8). We have shown that the scale has four generations of odontode deposition allowing for both partially and fully buried odontodes, while the spine tip bears only two generations of odontodes, with one partially buried generation, that are confined to the most posterior part of the lateral surfaces. Currently it is impossible to say whether the spine possesses more generations of odontodes basally, but this idea cannot be ruled out completely without further investigation.

Ridgelets that provide ornamentation on individual odontodes are similar between the scale and the spine. The features on youngest generation odontodes are rounded with the median ridgelet on each being wide and smooth, ending in a point (Figs. 3, 4 and 8). Older generations of odontodes share that same morphology, but have sharper features. The smoother ridgelets of the youngest generation odontodes are most likely due to erosion as both rounded and sharp ridges can be observed on the same odontode depending on whether the surface is buried. Ridgelets extend from the median ridgelet and surround the entire circumference of each odontode on the scales. The spine odontodes have a similar overall organization, but the ridgelets on buried surfaces have a more random distribution and do not become nodular proximally. The other spine odontodes do not preserve these nodules, probably because they are not fully buried and have been eroded.

The vascularization of the spine and the scale of Lophosteus have some similarities, but are generally quite different. Overall, the spine has a more complex vascular system than the scale. The spine tip has a multi-tiered vascular system for the bony base and several individual systems for each odontode. The scale also has bulbous and rounded vascular systems for each odontode (Figs. 5H and 5I), but there is no central system within the bone itself. Instead, the scale has several basal canals that exit the base. The spine tip also has a basal canal but it is difficult to comment any more on that without more proximal scan data. Both the spine and the scale have ascending vascular canals within each odontode. The ascending canals in the scale are attached to the bulbous pulp cavity of each odontode (Figs. 5H and 5I). This is also true for the spine, but these odontodes are not fed by bulbous pulp cavities (Fig. 10); instead, they are attached to the outermost bone vascular canals. However, the ascending canals in the spine are only open in the youngest odontodes, which are only located near the posterior margins of the spine and not evenly distributed over the entire surface (Fig. 10). A full scan of the Lophosteus spine is required to understand the odontode distribution and vascularization.

Both the spine and the scale have many vascular pore openings at the bone surface, as first suggested by Märss (1986) (Fig. 6). These can also be observed on the dentine depositional surfaces of buried odontodes. There is a second-generation odontode on the spine tip that has a pore opening on its surface connected to the ascending canals within it (Fig. 8H). This feature is unique to that particular odontode and cannot be observed elsewhere on the spine and is most likely the result of weathering.

Comparison of the scale and spine of Lophosteus to other taxa

Rohon (1893) described scales of Lophosteus for the first time and considered this taxon as a sarcopterygian. Gross (1969) identified Lophosteus as an early osteichthyan that cannot be assigned to either actinopterygians or sarcopterygians, based on the shape and histology of scales and spines. It is the third osteichthyan for which detailed 3D histological data have been obtained from the scales, the other two being Andreolepis (Qu et al., 2013) and Psarolepis (Qu et al., 2016).

The morphology of Lophosteus scales is consistent with known osteichthyan scales, with a rhomboid shape, peg-and-socket structure and an anterior overlapped field (Schultze, 2015). The lack of enamel on the scale surface of Lophosteus does not result from post-mortem erosion, as the embedded odontodes (which have not been subjected to post-mortem erosion) confirm the absence of an enamel layer on top of the dentine. Lophosteus is the only known osteichthyan that has a dermal odontode skeleton entirely devoid of enamel, a characteristic that supports its placement in the osteichthyan stem group (Qu et al., 2015b).

The scale crown is composed of four generations of odontodes. All figured scales of Lophosteus show each generation consisting of more than one odontode. The odontodes from a given generation are not in contact with each other, and their contact with the underlying odontodes of the previous generation is mediated by bone of attachment; there is no direct dentine-on-dentine contact. This is different from the pattern of Psarolepis scales and Andreolepis scales, in which the odontodes are in direct contact with each other by dentine, sometimes united by a shared enamel layer, and are added one at a time so that they cannot really be grouped into generations unless you consider each “generation” to contain just one odontode (Qu et al., 2013; Qu et al., 2016). The morphology of first generation odontodes in Lophosteus is similar to the first (primordial) odontode of Andreolepis scales (Qu et al., 2013), with a pointed posterior end and a slender triangular shape. However, Lophosteus odontodes have more ridgelets posteriorly and these ridgelets bear several nodules forming serrations (Fig. 4A), similar to younger odontodes. Although the most superficial odontodes are heavily eroded and such serrations become faint (Fig. 4D), all embedded odontodes show such serrations clearly (Figs. 4A–4C). The ridgelets on odontodes (either embedded or exposed) of Andreolepis and Psarolepis are smooth and have no such serrations with protruding nodules. On the other hand, such serration is common in some placoderm scales, such as Romundina (Ørvig, 1975). The nodules on serrations are delicate structures and can be easily destroyed by post-mortem erosion, making it difficult to evaluate the feature in other placoderms. Thus more placoderm scales (especially from articulated specimens) need to be scanned to reconstruct embedded odontodes.

The organization of the canal system in Lophosteus is similar to that of the Andreolepis scale. In the bony base there are three canals connected with a horizontal vascular network. All three basal canals are tilted anteriorly, similar to Andreolepis and Psarolepis scales. In addition, there are several isolated canals that do not connect with any other canals. No such isolated canals are found in Andreolepis or Psarolepis scales (Qu et al., 2016). The horizontal vascular canals are much thicker than those of Andreolepis and Psarolepis scales, especially for the bulbous canals under large young odontodes (Figs. 5D and 5E). The horizontal vascular canals are flattened (Fig. 5F) in basal view in all three taxa.

While there is a small amount of directly comparable synchrotron data (Qu et al., 2013; Qu et al., 2016) for the Lophosteus scale, there is no directly comparable 3D data yet published relating to the spine. Soler-Gijón (1999) provided a three-dimensional model of fin spine histology from a xenacanth shark to discuss the structure’s growth, but this was created from a stack of spaced 2D thin sections for which details can sometimes remain uncertain (e.g., osteostracans histology; Qu et al., 2015a). There are a large number of 2D histological descriptions of early vertebrate spines (Soler-Gijón, 1999; Jerve et al., 2014; Burrow et al., 2016) that can be compared to the thousands of virtual thin sections that comprise a synchrotron dataset, some of which we will briefly summarize here. However, it is not possible to comment in great detail on the morphology of structures that are only known from the 3D reconstructions (i.e., details relating to buried surfaces and individual odontode morphology, 3D architecture of the vascular canals).

The composition of the acanthodian and chondrichthyan fin spines is different from Lophosteus. Acanthodian fin spines (Climatius, Parexus) are composed of osteodentine and mesodentine, with some lacking cellular bone (Burrow et al., 2010; Burrow et al., 2013; Burrow et al., 2015) while others have it (Nostolepis; Denison, 1979). Fin spines of fossil and extant chondrichthyans differ even more from Lophosteus in that they lack bone altogether and are composed of different proportions of lamellar osteodentine and trabecular dentine that can be covered to varying degrees in mantle dentine and enameloid, depending on the taxon (Maisey, 1979; Jerve et al., 2014). In extant (and probably in fossil) chondrichthyans, the tips of the spines are shaped by an intitial epithelial fold, which defines the outer surface of the mantle dentine. This dentine grows centripetally, deposited by odontoblasts that differentiate from the mesenchyme contained within the epithelial fold; in effect the tipregion of the spine behaves like a single large odontode. The trunk dentine, which can comprise both lamellar and trabecular parts, develops within the mesenchyme of the spine primordium without contact with an epithelium (Maisey, 1979; Jerve et al., 2014). The histology of the Lophosteus fin spine tip shows compositional similarity with the dermal plates of acanthothoracid placoderm fish (Giles, Rücklin & Donoghue, 2013), but the vascularization in the tip of the spine suggests that the oldest part of the spine includes the tip. However, scan data taken from the midline and base of the spine are necessary to confirm this.

The odontode sculpture of Lophosteus is similar to the ornamentation found on acanthothoracid placoderms, like Romundina stellina (Ørvig, 1975). The ornamentation on most acanthodian spines appears to be more linear and continuous along the length of the spine (Miles, 1973), but there are some taxa that bear linear ridges that transform into nodules toward the base (Burrow et al., 2015). In the acanthodian Gyracanthides murrayi the pectoral fin spines have ornamentation where the individual nodules bear a stellate arrangement of ridges and somewhat resemble the ornament odontodes of Lophosteus and acanthothoracids (Warren et al., 2000). This ornament morphology was also reported by Miles (1973) to be present on the pectoral spines of the acanthodian Vernicomacanthus uncinatus. However, unlike in Lophosteus this ornament always seems to consist of a single layer, with no suggestion of multiple generations of odontode formation.

Some chondrichthyan fin spines share the acanthodian type of ornamentation with certain fossil sharks such as Asteracanthus and other hybodonts having a very thick layer of dentine ornament, but this has been greatly reduced in extant species like Heterodontus, Squalus, and Callorhinchus milii (Jerve et al., 2014). Tooth-like nodules are also present on acanthodian and chondrichthyan dorsal fin spines and are usually located on the most apical part of the posterior side, or trailing edge. In Gyracanthides murrayi and Callorhinchus milii, they are positioned in rows and are independent of any linear ridging and/or nodular ornamentation on the lateral sides of the spines (Warren et al., 2000; Jerve et al., 2014). This feature is not present on any of the Lophosteus spines. Lophosteus also differs from chondrichthyans and acanthodians in its distinct posterior spine surface and the extent of the attachment fibers (Sharpey’s fibers) in this area. This surface, which must have formed the attachment for the fin, extends all of the way to the spine apex, indicating that the spine did not have a projecting free tip like in Callorhinchus, Squalus, or acanthodians. Rather, the spine may have formed with the purpose of providing support for the leading edge of the fin.

Conclusions

The description of the scale and spine tip of Lophosteus presented here shows that there is a great deal of histological information that can be derived from high-resolution 3D datasets. Not only are we able to confirm the previously published characteristics from this taxon by Gross (1969) and others, but we have also shown that 3D synchrotron data can aid in identifying new morphological and potentially important paleohistological features. The following are the main novel features uncovered by the study:

• The spatial organization of dermal odontode addition on the scale and spine. These are broadly similar, both based on a gap-filling system and lacking the type of unidirectional ‘dentition-like’ addition seen on the scale of Andreolepis (Qu et al., 2013), but the scale shows a greater number of odontode generations.

• The morphology of the early-generation buried odontodes, including fine details of their surface ornament. This proves to be important from a taxonomic perspective, because the buried odontodes on the scale have a morphology identical to that used by Schultze & Märss (2004) to define the new species Lophosteus ohesaarensis. It seems probable that this species is based on ontogenetically young material of L. superbus.

• The 3D organization of the pulp cavities and vasculature of the scale and spine, including the presence in the spine of several distinct but connected vascular canal systems with different geometries.

• The presence of a complex mesh of Sharpey’s fibres in the posterior face of the spine, probably indicating the attachment of the associated fin.

The large number of histological similarities between the scale and the spine imply that these are characteristics that could be used for phylogenetic analysis as well as studying biological processes and development. Because of the current limited availability of comparable 3D data (Qu et al., 2013; Qu et al., 2016), we have elected not to attempt a phylogenetic investigation or definition of discrete characters at this stage, but we are confident that the continuing rapid expansion of this data set with the description of new early vertebrate histologies will eventually have a profound impact on our understanding of deep vertebrate interrelationships.

Our sincerest gratitude goes to Paul Tafforeau (ESRF) for facilitating the synchrotron beamtime provided by the ESRF. We would also like to acknowledge Tiiu Märss (Institute of Geology at the Tallinn University of Technology, Estonia), for access to material and to the field site on Saaremaa. Many thanks to Barbro Bornsäter Mellbin and Vincent Dupret (Australian National University) for scanning the material, and Tatjana Haitina (Uppsala University), Daniel Snitting (Uppsala University), Michael D Gottfried (Michigan State University) for all technical, administrative, and editorial support. Thank you to Git Klintvik (Lund University, Sweden) for acid preparations of the samples. Many thanks to Min Zhu, John Long, and Hans-Peter Schultze for providing reviews of this work.

Abbreviations

al anterior ledge

adc ascending canal

aof anterior overlapped field

bc basal canal

bco basal canal opening

bco.n basal canal opening, not connected to the horizontal vasculature

bvc bone vascular canal

cvc central vascular canal

den denteon

dt dentine tubules

dvc dentine vascular canal

eo embedded odontode

fgo first generation odontode

g (1–4) generation(s) (1–4)

k keel

leo leading edge odontode

med.c median canal

o osteocyte lacuna

pl posterior ledge

ps posterior surface

s sediment

sgo second generation odontode

shf Sharpey’s fibers

v void space/pseudocanal?

vc vascular canal

vco vascular canal opening/pore opening

Additional Information and Declarations

Competing Interests

Author Contributions

Data Availability

The authors declare there are no competing interests.

Anna Jerve performed the experiments, analyzed the data, wrote the paper, prepared figures and/or tables, reviewed drafts of the paper.

Qingming Qu conceived and designed the experiments, performed the experiments, analyzed the data, wrote the paper, prepared figures and/or tables, reviewed drafts of the paper.

Sophie Sanchez conceived and designed the experiments, performed the experiments, contributed reagents/materials/analysis tools, wrote the paper, reviewed drafts of the paper.

Henning Blom conceived and designed the experiments, reviewed drafts of the paper.

Per Erik Ahlberg conceived and designed the experiments, wrote the paper, reviewed drafts of the paper.

The following information was supplied regarding data availability:

European Synchrotron Radiation Facility.

The data are available on the ESRF Paleontology database: http://paleo.esrf.eu.

This information has been included in the body of the manuscript.

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
