# Peer review of "Three-dimensional paleohistology of the scale and median fin spine of Lophosteus superbus (Pander 1856)"

_PeerJ, doi:10.7717/peerj.2521_

## Round 0.1 · original submission · Minor Revisions

Please address all of the referees' suggestions, especially the detailed comments by Reviewer 2, in preparing a revised version of the manuscript.

·

Basic reporting

Lophosteus is an enigmatic and debatable taxon in the quest of the origin and early evolution of osteichthyans or bony vertebrates. This well-written manuscript provides the first description of the morphology and paleohistology of a fin spine and a scale from Lophosteus using virtual thin sections and 3D reconstructions, and adds a lot to the understanding of the nature of this key taxon along the stem lineage of osteichthyans. The figures are in high quality and well described.

Experimental design

The discussions and conclusions are objective, logically consistent, and rigorous, based on the available data. Methods in this manuscript are described with sufficient detail. In summary, this is an important contribution to the research of early vertebrates and paleohistology, with its original primary research within the PeerJ scope.

Validity of the findings

The description of the two scans and restorations is detailed, informative and clear, thus providing the rich source of morphological and paleohistological data for the further discussions relating to early vertebrate growth and phylogeny.

Additional comments

My comments and revisions are minor, mainly relating to some typo errors or few points to be clarified. They are annotated in the attached pdf file for the reference of the editor and the authors.

·

Basic reporting

Clear, unambiguous, professional English language used throughout. yes

Literature: citations missing, citations wrong, papers in references, which are not cited. And relevant citation missing.

Figures: can be reduced (e.g. fig. 4 is a repetition of fig. 3B); 3D figures high quality, 2D figures grey, bad quality (differences between tissues not visible)

Experimental design

The authors refer to new information by 3D synchrotron data, that is only the case for the morphology and the canal system, the histology is still 2D and with bad, grey pictures (less informative than Gross 1969)

Validity of the findings

not very new, repetition of Gross 1969
data are robust

Additional comments

Title misleading, there is only a Two-dimensional paleohistology .. nevertheless a Three-dimensional morphology of external features and the internal canal system ….
All the virtual sections are 2D not 3D; they are grey and do not show the tissues (figures by Gross 1969 are much more informative). You need to show thin sections to justify denteons.
Literature: citations missing, citations wrong, papers in references, which are not cited. And relevant citation missing.

An annotated document is attached - the red text shows all my comments and annotations.

·

Basic reporting

No comments

Experimental design

No comments

Validity of the findings

No comments

Additional comments

This is a solid descriptive paper presenting important new information on the histology and microstructure of an important fossil taxon, Lophosteus. Due to its unresolved phylogenetic position, any new data on this genus is vitally important for helping place it phylogenetically and to make meaningful comparisons with other 'scale and part' taxa known principally from fragmentary remains.
The descriptions are good, as are the illustrations. I would like to see a short summary in the conclusions listing precisely what new features/histological structures have been found in this paper, to be clearly laid out for readers who are not familiar with this topic. Other than that I have no real criticism of the work, it is very detailed and could be published as is. My suggestion is only to improve the final version for readers outside this specialised field.

---

## Round 0.2 · accepted · Accept

The authors have thoughtfully evaluated and responded to each of the referees' comments, and the AE finds the responses appropriate. The AE is pleased to recommend the revised manuscript for acceptance for publication.